# Sexual Knowledge, Attitudes and Behaviours among Undergraduate Students in China—Implications for Sex Education

**DOI:** 10.3390/ijerph17186716

**Published:** 2020-09-15

**Authors:** Jinping Lyu, Xiaoyun Shen, Therese Hesketh

**Affiliations:** 1Center for Global Health, School of Public Health, Zhejiang University, Hangzhou 310058, China; norahlv@zju.edu.cn; 2The Institute for Global Health, University College London, London WC1N 1EH, UK

**Keywords:** sexual knowledge, sexual behavior, students, homosexuality, gender differences, sex education, China

## Abstract

Background: Despite rapid modernization and improving gender equity in China in recent decades, traditional values prevail in many areas of life, including sexual behavior. This study aimed to explore gender differences in sexual knowledge, sexual attitudes and sexual behaviors, as well as preferences for sex education among undergraduates in China. Methods: A cross-sectional study surveyed 5965 undergraduates (62.8% females), aged 15 to 24 years from nine universities in Zhejiang, Henan and Yunnan provinces, from September to November 2019. Results: Of the total sample, 158 (2.6%) self-identified as homosexual, 287 (4.8%) as bisexual and 324 (5.4%) stated they were unclear about their sexual identity. The mean sexual knowledge score out of 12 was 6.16 ± 2.54 points. Ever having sexual intercourse was reported by 18.7% (27.0% males, and 13.9% females). Students from urban backgrounds, and those with homosexual and bisexual orientation were more likely to have had sexual intercourse. Most students (72.5%) reported that they would prefer to receive sex education from on-line sources. Conclusions: Female students are significantly more conservative in sexual attitudes and sexual behaviors. Low levels of sexual knowledge contribute to risk behaviors among Chinese adolescents. China needs to develop and widely disseminate on-line sex education, with practical, age-appropriate content.

## 1. Introduction

China has the second largest number of adolescents in the world with an estimated 230 million of its population in the 10–24 years age group [1]. China’s rapid socioeconomic transition and opening-up to the outside world over the past three decades has had a huge impact on the lives of these adolescents. Educational opportunities expanded largely with 30.3 million undergraduates enrolled in 2019, up from 5.6 million in 2000 [2,3]. The internet and social media exposed adolescents [4], in particular, to globalized culture, and different social norms and values related to sex and marriage. These contrast sharply with the norms and values of traditional Chinese culture, which disapproves of sex outside marriage, especially for females [5,6].

This rapid societal change contributed to impacts on sexual and reproductive health outcomes. First, premarital pregnancy is a concern. In much of China, having a child outside marriage is still regarded as a violation of family planning regulations, sometimes leading to penalties and fines. As a result, such pregnancies usually result in abortion [7]. Accurate figures for premarital pregnancy are hard to obtain. A national survey, reported 63 pregnancies per 1000, in unmarried women aged 15–24 years old, with 82.5% ending in abortion [8]. A study among never-married sexually active females aged 15–24 years in Shanghai indicated that 13.8% had had an unintended pregnancy, with 99.0% ending in abortion [9]. Second, there is a recent increase in the incidence of HIV/AIDS in this age group, specifically among university students [10,11]. According to the National Center for AIDS/STD Control and Prevention, from 2011 to 2015, the number of university students infected with HIV increased by 35% annually, reaching 3236 in 2015 [12]. In 2017, 3077 (81.8%) of new diagnoses among university students were in men who have sex with men [13]. This occurred against the background of gradually increasing acceptance of homosexuality over recent years, though it is still generally stigmatized, especially in the family setting [14]. The authorities now address issues of sexuality, especially homosexuality, in more tolerant and less punitive ways [15]. However, being LGBT is still considered a form of shame for one’s family (‘losing face’) [16]. In contemporary China, strong traditional values co-exist with very liberal views about homosexuality [15].

There is good evidence from many countries that comprehensive sex education programs lead to safer sexual behaviors [17]. However, the implementation of sex education in China is still inadequate. In a 2017 survey of adolescent sexual and reproductive health knowledge, the average score for correct answers was 53% [18]. Another study among university students who had had sex found that 18% did not know how to use condoms and 44% did not know how to use birth control pills [19]. A study from 2008 of 591 teenagers seeking abortions in Shanghai, showed that 52% of them had never heard of emergency contraception [20]. Another study from 2009 showed that only 19.8% of Chinese university students knew the anatomical names for male genital organs [21]. These all indicate that the sexual knowledge among university students is still at a low level. Many young people turn to the Internet, including pornography, for information as a resource for sexual knowledge [22].

Sexual attitudes and behaviors are changing in China. As in many countries, the shift towards later marriage has led to an increase in premarital sex [23], and premarital sex has become widely accepted among young people [24]. An early study found rates of premarital sex among Chinese university students in 1989 to be 13% for men and 6% for women, increasing to 19% for men and 17% for women in 1993 [24]. In the 21st century, one study in Shantou showed 76% of university students considered pre-marital sex to be acceptable, and 21% had had sexual intercourse [25]. In another study of 2071 female university students from urban areas, 57% had acceptable and neutral attitudes towards premarital sex [22]. Of 1355 adolescents surveyed in 2017, 74% said it was acceptable to have sex before marriage [18]. In addition, attitudes towards homosexuality appear to be shifting. A study in 2002 in Changsha, found that 37% of students said homosexuality should be illegal and 35% would stop being friendly with someone they discovered was homosexual [26]. In 2017, 34% of 1355 adolescents believed homosexuality is a normal sexual orientation, but 11% believed that homosexuality is abnormal [18]. In another study of 2071 female students, 80% had accepted and neutral attitudes towards homosexuality [22].

Another emerging tension relates to changes in gender equality over recent years. China has been a patriarchal society for thousands of years. According to Confucianism, the virtue of a woman lies in the three obediences—obedience to the father, husband and son [27]. During the Mao years from 1949, much progress towards equality was made. Concubinage and prostitution were forbidden, divorce became easier to obtain and women were encouraged to work outside the home [27]. However, Confucian tradition in gender roles lingers. For example, women with childcare responsibilities find it more difficult to get jobs in many sectors and society is still more tolerant of pre-marital and extramarital sexual activity in males [28]. Females appear to be tolerant of infidelity in their partners and even their use of commercial sex, while men demand and expect fidelity [15]. There remain power imbalances between males and females in sexual decision-making, which makes it particularly urgent for females to obtain more sexual knowledge. Gender differences are also manifest behaviors. Studies found that male students are more curious about sex [29] and they watch more pornography [30]. Several studies showed that male university students are more inclined to have premarital sex than female students [25,29,30]. Concerns about pregnancy might cause female university students to be more cautious about sexual intercourse [29].

All of the above have led to calls for better sex education for adolescents. Initial awareness of the need to understand more about sexual knowledge and attitudes among young people was largely driven by the emergence of HIV in the 1990s. According to current Chinese national guidelines on health education in primary and middle schools, sex education should provide age-appropriate educational content, but emphasize on reproductive system diseases and sexual morality, with warnings about “the negative influences of pre-marital sex” [31]. In 2011, an addition was made, which stated that university students should receive courses on the “psychology of sex and love” [32]. In 2017, it was recommended that elective courses on sex education be added to university health education [33]. However, school and university-based sex education programs are not formally evaluated, so there is no incentive to deliver such programs [34]. A recent survey indicated that only about 55.6% of 17,966 undergraduates received any kind of sex education, nearly all of which focused on the biology of sex [34,35]. In addition, attention should be paid to gender differences in sex education. Men are more likely to instigate sexual activity [35], so sexual education needs to have a focus on gender equity. This includes empowering women and giving them the confidence to refuse sex, if they do not want it, while men need to learn to respect women. A study of students found that boys’ scores for attitudes on gender equality improved significantly, after sex education made them aware of it [36].

Open discussion about sex in mainland China is still largely a taboo and research in this area is still relatively new. Previous studies tended to focus on HIV infection and sexual behaviour [30,37], but this was not sufficient. The sexual situation of post-00s students has definitely changed, and as more LGBT individuals are willing to reveal their sexual orientation, sex education needs to include this. This study aimed to understand current sexual knowledge, attitudes and behaviours, among undergraduate students in contemporary China and to identify the needs and preferences in sex education to inform policy. The study was designed to address three primary research questions—first, what is the current status of sexual knowledge, attitudes and behaviours of university students (including LGBT students) in China today? Second, what are the gender differences in sexual attitudes and sexual behaviours? Finally, what are the needs of university students for sex education?

## 2. Materials and Methods

### 2.1. Study Design and Participant

We carried-out a cross-sectional study, consisting of an on-line survey, in Yunnan, Henan and Zhejiang Provinces. Research about Chinese university students’ sexual knowledge, attitudes and behaviors mostly focused on one university, or on one geographical area [30,37]. Our selection of provinces was aimed to be more representative, by including provinces from the west, central and eastern regions of China and representing low, medium and high levels of economic development, respectively. The sample also aimed to broadly represent the range of different levels of universities in China, so in each province, one university from each of the three levels (top-tier, first-tier and second-tier universities) was included.

### 2.2. Procedures

Data collection was carried-out from September to November 2019. At each of the nine universities, three research assistants were recruited. A unique link to the QR code for the questionnaire was sent to each assistant. The assistants then disseminated the link to undergraduates with the aim of collecting data broadly representative of different grades and majors. The data were collected through the Chinese professional survey website Wenjuanxing (www.sojump.com).

### 2.3. Measures

The questionnaire was divided into five broad sections, as follows—(1) sociodemographic, (2) sexual knowledge, (3) sexual attitudes, (4) sex-related behaviors and (5) sex education. The questions on sexual knowledge and behaviors were adapted from the World Health Organization Illustrative Questionnaire for Interview–Surveys with Young People (IQISYP), which is designed for unmarried teenagers and young people [38].

Sexual knowledge was assessed with 12 questions (11 single-choice questions and 1 multiple-choice question). Each single-choice item was measured through a true, false and not sure format. The multiple-choice question was to correctly identify transmission modes of HIV/AIDS from—contaminated needles, mosquito bites, from mother to baby, hugging, kissing, vaginal sex, anal sex, shared toothbrush and shared bathroom. The maximum score attainable was 12 points.

There were 22 questions on sexual attitudes; 15 questions drew on personal aspects drawn from the IQISYP. The remaining 7 questions addressed general aspects and came from the Attitudes Towards Sexuality Scale (Cronbach-alpha of 0.75 for adolescents, and the test-retest reliability of 18–20-year-old was r = 0.90 [39]). There were three options for each question—agree, neutral and disagree.

Sexual behavior items were adapted from the IQISYP and included questions about sexual experiences, relationships, use of contraception, pregnancy and masturbation.

Questions on sex education included perceived need and preference for content and modes of learning.

### 2.4. Ethical Considerations

Study participation was voluntary. Anonymity and confidentiality were assured. Ethical approval was obtained from the Medical Ethics Committee of Zhejiang University, School of Public Health (ZGL201904-5).

### 2.5. Statistical Analysis

The data were analyzed descriptively and then using Chi-squared test, *t*-tests, Analysis of Variance (ANOVA) and binary logistic regression analyses, while controlling for key variables. A *p*-value of less than 0.05 was deemed to be statistically significant. Data were analyzed using the Statistical Package for the Social Sciences (SPSS) version 21.0 (IBM Corp, New York, NY, USA).

## 3. Results

### 3.1. Socio-Demographic Characteristics of the Respondents

A total of 6665 questionnaires were received. After checking for completion of key variables and credibility of responses, 5965 (89.5%) were included in the analysis. As shown in Table 1, the mean age was 19.9 ± 1.46 years (range 15–24 years), 3747 (62.8%) were female, 3118 (52.3%) were from rural backgrounds and 2302 (38.6%) were only children. A total of 4340 (72.8%) students were brought up by both parents, 819 (13.7%) students by grandparents and 736 (12.3%) by a single parent. The highest level of education of most parents, 3707 (62.1%), was middle school.

### 3.2. Sexuality

Number of students who self-identified as heterosexual, homosexual, bisexual or unclear were 5196 (87.1%), 158 (2.6%), 287 (4.8%) and 324 (5.4%), respectively. Male students reported higher rates of homosexuality (3.9% vs. 1.9%), but lower rates of bisexuality (3.2% vs. 5.7%) and unclear orientation (3.7% vs. 6.4%) than females in Table 2. Students from urban areas were twice as likely to self-identify as bisexual (6.7% vs. 3.1%). Homosexuals and bisexuals were more likely to have ever had sexual intercourse (34.2% and 25.1% vs. 18.7% for heterosexuals).

### 3.3. Sex-Related Knowledge

The mean knowledge score out of a total of 12 was 6.16 ± 2.54 points; 28 students (0.5%) gave correct answers to all 12 questions. In terms of pregnancy, 3649 (61.2%) thought it was possible for a woman to get pregnant the first time she had sex, and 48.8% had an understanding of the “safe period” (with 33.4% unsure). Around one-third of students correctly recognized all transmission modes for HIV and half (50.9%) knew there were simple tests for diagnosis of HIV; 4978 (83.5%) students knew that condoms protect against STIs and 5238 (87.8%) thought that condoms could be used more than once. However, 2400 (40.3%) students thought that masturbation was harmful to health (with 26.2% unsure). Across all items, an average 22.6% of respondents answered ‘not sure’ (Table 3).

Sex-related knowledge scores were significantly higher in males, in higher grades (older students), universities in the Zhejiang province, in top-tier universities, science majors, urban students, those with homosexual or bisexual orientation and those with sexual experience (Table 4).

### 3.4. Sexual Attitudes

Across the range of sexual attitudes, there were significant differences between the sexes for most items (Table 5). In terms of attitudes to personal sexual behaviours, male students were more likely to think there is nothing wrong with premarital sex (50.4% vs. 41.1%), that sex is acceptable if methods are used to stop pregnancy (45.0% vs. 36.7%), that one night stands are acceptable (23.5% vs. 9.3%) and that it is the girl’s responsibility to ensure contraception is used (13.7% vs. 8.8%). Importantly, 8.7% of boys thought it was acceptable to force a girl to have sex. Female students were more likely to agree that people should be in love before having sex (68.1% vs. 59.3%).

Regarding general sex-related attitudes, female students were much more likely to agree that homosexuality is acceptable (61.5% vs. 39.9%). Males were more likely to agree that a person who catches a sexually transmitted disease “is getting exactly what he/she deserves” (19.5% vs. 8.6%). Access to abortion on demand was supported by 18.2% of males and 27.5% of females.

### 3.5. Sexual Behaviors

Of the 5965 students, 1118 (18.7%) students reported ever having sexual intercourse, with males being twice as likely to have had sex (27.0% vs. 13.9%). Of these, males were also more likely to report having more than one partner (41.8% vs. 28.8%). Of those reporting ever having sex, 18.2% of males and 11.9% of females reported having first sex under the age of 18 years. In terms of first sexual intercourse, 21 (4.0%) female students reported they were forced to have it, 16 males admitted forcing their partner and 26.1% students reported that they did not use any contraceptive methods when they first had sex. The main reasons given for not using contraceptives were that they did not prepare for them (65.1%) and did not know how to use them 10.3%.

Ever having a boyfriend or girlfriend, was reported by 56.1% (59.3% males, 54.1% females) and 68.3% (57.3% males, 75.4% females) had not had sex with them. Of those currently in a relationship, males were more likely to report having sex than females (50.6% vs. 35.3%), and the overwhelming majority (84.9% males, 90.1% females) said they were in a serious relationship, likely to lead to marriage. Ninety-three students, 59 males and 34 females, reported that they were involved in a pregnancy in their current relationship, 39 had been aborted, 19 were still pregnant and there were nine births (Table 6).

Table 7 shows the associations with having had sexual intercourse—male sex (OR = 2.38, 95% CI (2.07–2.74), *p* < 0.001), higher grade/age (OR = 2.70, 95% CI (2.22–3.30), *p* < 0.001 for Years 4–5, compared with Year 1), urban background (OR = 1.42, 95% CI (1.23–1.63), *p* < 0.001) and homosexual (OR = 2.12, 95% CI (1.50–3.01), *p* <0.001) and bisexual orientation (OR = 1.65, 95% CI (1.24–2.20), *p* = 0.007).

There was a large sex difference in masturbation behaviors—71.5% of males and 27.1% of females (*p* < 0.001) reported ever masturbating, with frequencies several times a day, once a day, several times a week, month and year—5.5% (6.4% males, 3.9% females), 5.3% (6.1% males, 4.0% females), 22.5% (26.9% males, 15.5% females), 41.8% (42.9% males, 40.0% females) and 25.0% (17.6% males, 36.5% females), respectively.

### 3.6. Sex Education

Most students wanted sex education in all the areas that are listed in the questionnaire—sexual and reproductive systems (73.6%), relationships including LGBT-Q (70.1%), puberty (59.6%), STDs (59.1%) and contraceptive methods (54.0%). Three-quarters wanted to learn from the internet, 68.5% from courses provided by sexual health experts (including on-line), 33.8% through sexual experience and 28.6% from friends.

## 4. Discussions

Our study highlights a number of the issues facing young Chinese today, as China undergoes a social and cultural transition. In particular, our findings highlight the need for appropriate sex education in early adolescence. A number of specific issues emerge.

First, among university students, attitudes and behaviour with regard to sex are mostly conservative in the context of a modern society. Just over half (59% of boys and 54% of girls) had ever had a romantic relationship and 27% of boys and 14% of girls had ever had sex. While low compared to undergraduate students in many other countries [40,41,42], sexual activity appears to be gradually increasing among adolescents in China [29]. Higgins et al. found that 97% of university students in Beijing lacked any sexual experience two decades ago [28]. A meta-analysis showed that the overall rate of ever having sexual intercourse was 15% from 2005 to 2009, and 17% from 2010 to 2015 among mainland Chinese university students, with 21% among males and 11% among females, across the whole period [29]. Many universities in China have direct or indirect regulations to limit intimacy between students of the opposite sex. For example, single sex dormitory buildings prohibit access to students of the opposite sex and shared dormitories prevent privacy. For this reason, hotels near campuses are known to be used for sexual activity.

Second, while sexual activity is low, failure to use contraception leading to unplanned pregnancy, presents a problem. A total of 26% of respondents did not use contraception during first sexual intercourse. Of students currently in a relationship, 93 (12%) students reported a pregnancy, and 39 (42%) ended in abortion. Of the live births, all were reported by male students. What exactly happened to these babies was unclear as we did not ask about the outcomes. Our pregnancy rate was considerably lower than in another study, which showed the prevalence of unintended pregnancy in 3595 sexually active female university students as 32%, with an abortion rate of over 80% [7]. The authors explained this as related to the shame still attached to premarital pregnancy and very easy access to abortion [7].

Thirdly, there are distinct differences in the sexual attitudes between males and females, though overall attitudes across both sexes are relatively conservative. Female students were less accepting of premarital sex and one-night stands, but were more accepting of homosexuality. Another study showed both male and female undergraduates believed that premarital sex is more acceptable for males than for females [43]. In Chinese society, in general, premarital sexual activity in males is regarded with more tolerance than premarital sexual activity in females [30]. This might explain gender differences in sexual intercourse in this study. It is considered to be a rite of passage for boys, while it leads to girls being labelled and stigmatized [30]. Men continue to have more respect and privileges than women [44]. The findings combine to show that traditional values of sex are still deep-rooted in the Chinese society.

Fourth, more than 50% of students (62% female, 40% male) have an accepting attitude towards homosexuality. Attitudes towards homosexuality have changed greatly in the past two decades. In this study, 13% of students reported they were not heterosexual. This is twice the proportion of non-heterosexual youth reported in a study from 2009 [45]. Only in 2001, homosexuality was removed from the Chinese Classification of Mental Disorders [46]. In China, sexual surveys still often do not ask about sexual orientation [30,37]. The attitudes towards homosexuality in our study are in sharp contrast to the findings of 2006 survey among 6299 Shanghainese students aged 15–24 years, which showed only 20% of males and 28% of females thought homosexuality was acceptable [47]. Another study found that young and well-educated people were more becoming more tolerant of homosexuality [48]. Unsurprisingly, the evidence suggests that contemporary university-educated adolescents are at the forefront of growing tolerance towards homosexuality. In addition, homosexual and bisexual students had higher levels of sexual knowledge and were more likely to have had sexual intercourse than heterosexuals. In a national survey conducted in 2000, less than 3% of adults aged 20–64 years reported that they had had a same-sex sexual relationship [49]. The expanding HIV epidemic among men who have sex with men (MSM) in China has forced LGBT health issues into the open [46]. In 2017, a national report, showed that homosexual transmission accounted for 26% of all new cases of HIV, and among students, 82% were homosexually transmitted [13]. In one study in students from Nanjing, 95% of 156 positive HIV tests were from MSM [50]. There is a clear need for appropriate sexual and reproductive health services for MSM. However, persisting stigma that is strongly felt by LGBT individuals remains a considerable a barrier.

Fifth, responses showed that 194 (9%) males believed that forcing a partner to have sex is acceptable and 101 (3%) of females agreed with this. Asked about first sexual experience, 16 males admitted to forcing their partner and six girls said they were forced. One study in Shanghai among 1099 undergraduates showed 2% and 5% of boys and girls, respectively, had encountered forced sexual intercourse [43]. The authors of this study suggested that force in the context of a committed relationship might not be interpreted as “forced” sex and actual rates might therefore be higher [51], which might mean that our figures also represent an underestimate.

Finally, sexual knowledge was generally low and this clearly needed to be addressed, as was noted in other studies among Chinese university students [52]. Our respondents averaged 50% on the knowledge test. However, in this study, we found that many students responded “Not sure” to several questions, demonstrating their low level of knowledge. This, of course, raises questions about sex education. There is no national curriculum for sexuality education in China [53]. In a previous study, nearly half of the undergraduates reported never receiving school-based sex education [35].

Our respondents’ preference for teaching approach was on the internet. In the post Covid-19 era, internet learning has become the norm, making it a particularly desirable way of teaching sensitive and potentially embarrassing topics. There are a number of trustworthy websites providing sex, sexuality, or STI-related information in English. However, the Chinese websites are often of poor quality [23]. For example, the Planned Parenthood Federation of America has a sex education website, specifically aimed at teens, with topics including—sex, puberty, preventing STDs and pregnancy, relationships and LGBTQ [54]. On this website, for example, the topic, ‘*How Do I know if I’m Pregnant*’ explains clearly what it means to be pregnant, how to do pregnancy tests, and what the options are for managing it and preventing future pregnancy. This is the sort of material that is really necessary for Chinese adolescents. Universities can be a good source of sexual education. For example, Columbia University has set up a website, ‘*Go Ask Alice*’ to assist university students with sexual health issues [55]. The official website of the China Family Planning Association, which is responsible for the youth health university program does not even have a section on sexual knowledge. A few Chinese Civil Society Organisations (CSOs) began to establish online sex education platforms. Among them, the sex education support platform called Niwohuoban developed courses in sex education for school and university students [56]. The website is mainly aimed at schools, allowing them to use teaching materials free of charge to teach students sex education. However, the resources are limited and do not meet practical needs of adolescents in terms of sexual behaviour, such as protection against STIs and pregnancy, how to use condoms, issues of consent and LGBTQ. China needs to develop an on-line sex education programme, with practical, age-appropriate content across the age range from primary school through to universities. This needs to be developed by pedagogic experts working with young people, to ensure the suitability of content. Best resources from overseas should be sought, then adapted, edited and translated for the Chinese setting. The programme should be modular allowing students to access materials that meet their own needs. Such a programme needs official approval in China, but once obtained, the programme can then be made rapidly accessible across the whole country.

Sex education can also play an important role in helping to address gender inequity. There needs to be a focus on empowering women, while men need to understand the importance of respecting women, to be aware of the potential power imbalance. A review compared two types of sexuality education intervention programs across 22 studies in the United States, the United Kingdom, China, South Africa and other countries. One type had specific gender equality content, while the other did not. The former was much more effective in decreasing the rate of unintended pregnancy and sexually transmitted diseases [57]. Through sex education, awareness of equality between males and females could be raised, leading to reduced sexual coercion and helping to reduce unintended pregnancy and abortion.

Limitations of this study should be noted. First, there were more female respondents, though this is a common form of selection bias in on-line voluntary surveys. Second, while our participants were recruited from three provinces, from the eastern, central and western regions, the findings were not generalizable to undergraduates across the country. Furthermore, the reporting bias is a problem for all self-completion surveys and is a particular problem when addressing personal and sensitive topics.

## 5. Conclusions

Relatively low levels of sexual knowledge among Chinese adolescents contribute to unsafe sexual behavior. This makes the development of age-appropriate, high quality, easily accessible, on-line sex education for the whole of China, a priority. A gender-sensitive approach to sex education should be emphasized, with a focus on empowering girls.

## Figures and Tables

**Table 1 ijerph-17-06716-t001:** Socio-demographic characteristics of respondents.

Characteristic	Total (SD/%) *N* = 5965	Male (%) *N* = 2218	Female (%) *N* = 3747	χ^2^	*p* Value
**Mean age**	19.89 ± 1.46	19.90 ± 1.52	19.89 ± 1.42	0.17	0.87
Grade					
1st Year	1583 (26.5)	761 (34.3)	822 (21.9)	113.14	<0.001
2nd Year	1753 (29.4)	584 (26.3)	1169 (31.2)		
3rd Year	1412 (23.7)	445 (20.1)	967 (25.8)		
4th–5th year	1217 (20.4)	428 (19.3)	789 (21.1)		
**Location of Universities**					
Yunnan Province	1394 (23.4)	435 (19.6)	959 (25.6)	60.72	<0.001
Henan Province	2965 (49.7)	1246 (56.2)	1719 (45.9)		
Zhejiang Province	1606 (26.9)	537 (24.2)	1069 (28.5)		
**School Category**					
Top-Tier Universities	2352 (39.4)	1024 (46.2)	1328 (35.4)	162.01	<0.001
First-Tier Universities	2537 (42.5)	709 (32.0)	1828 (48.8)		
Second-Tier Universities	1076 (18.0)	485 (21.9)	591 (15.8)		
**Majors**					
Science	2524 (42.3)	1185 (53.4)	1339 (35.7)	178.65	<0.001
Arts	3441 (57.7)	1033 (46.6)	2408 (64.3)		
**Childhood Residence**					
Urban	2847 (47.7)	1095 (49.4)	1752 (46.8)	3.81	0.051
Rural	3118 (52.3)	1123 (50.6)	1995 (53.2)		
**Only Child**					
Yes	2302 (38.6)	1043 (47.0)	1259 (33.6)	105.95	<0.001
No	3663 (61.4)	1175 (53.0)	2488 (66.4)		
**Main Caregiver**					
Grandparents	819 (13.7)	299 (13.5)	520 (13.9)	12.64	0.006
Both Parents	4340 (72.8)	1659 (74.8)	2681 (71.6)		
Single Parent	736 (12.3)	243 (11.0)	493 (13.2)		
Others	70 (1.2)	17 (0.8)	53 (1.4)		
**Family Income Level**					
Below Average	643 (10.8)	306 (13.8)	337 (9.0)	34.34	<0.001
On Average	3488 (58.5)	1237 (55.8)	2251 (60.1)		
Above Average	1834 (30.7)	675 (30.4)	1159 (30.9)		
**The Highest Level of Education of one Parent**					
Below Primary School	229 (3.8)	90 (4.1)	139 (3.7)	10.66	0.014
Primary School	535 (9.0)	197 (8.9)	338 (9.0)		
Middle School	3707 (62.1)	1326 (59.8)	2381 (63.5)		
University and Above	1494 (25.0)	605 (27.3)	889 (23.7)		

**Table 2 ijerph-17-06716-t002:** Respondents by sexual orientation.

Characteristic	Heterosexual (%)	Homosexual (%)	Bisexual (%)	Unclear (%)	*T* or χ^2^	*p* Value
*N* = 5965	5196 (87.1)	158 (2.6)	287 (4.8)	324 (5.4)	-	-
Gender						
Male *n* = 2218	1976 (89.1)	87 (3.9)	72 (3.2)	83 (3.7)	59.75	<0.0001
Female *n* = 3747	3220 (85.9)	71 (1.9)	215 (5.7)	241 (6.4)		
Childhood Residence						
Urban *n* = 2847	2402 (84.4)	82 (2.9)	191 (6.7)	172 (6.0)	50.27	<0.0001
Rural *n* = 3118	2794 (89.6)	76 (2.4)	96 (3.1)	152 (4.9)		
Ever Had Sex						
Yes	972 (18.7)	54 (34.2)	72 (25.1)	20 (6.2)	65.92	<0.0001
No	4224 (81.3)	104 (65.8)	215 (74.9)	304 (93.8)		

**Table 3 ijerph-17-06716-t003:** Sex-related knowledge of Chinese university students by gender.

Statements	Total (%)		Male (%)		Female (%)		χ^2^	*p* Value
	Correct	Not Sure	Correct	Not Sure	Correct	Not Sure		
A woman can get pregnant on the very first time that she has sexual intercourse.	3649 (61.2)	1409 (23.6)	1368 (61.7)	504 (22.7)	2281 (60.9)	905 (24.2)	0.38	0.54
A girl does not grow much after menarche.	1651 (27.7)	2373 (39.8)	637 (28.7)	1076 (48.5)	1014 (27.1)	1297 (34.6)	1.91	0.17
Masturbation causes harm to health.	2001 (33.5)	1564 (26.2)	708 (31.9)	425 (27.2)	1293 (34.5)	1193 (30.4)	4.18	0.041
A woman is unlikely to get pregnant if she has sexual intercourse halfway between her periods.	2909 (48.8)	1993 (33.4)	1132 (51.0)	688 (31.0)	1777 (47.4)	1305 (34.8)	7.28	0.007
Unplanned or undesired pregnancies have a greater likelihood of miscarrying than do planned pregnancies.	2906 (48.7)	2088 (35.0)	1152 (51.9)	744 (33.5)	1754 (46.8)	1344 (35.9)	14.66	<0.001
Condoms are an effective way of protecting against sexually transmitted diseases.	4978 (83.5)	566 (9.5)	1884 (84.9)	197 (8.9)	3094 (82.6)	369 (9.8)	5.66	0.017
Male condoms can be used more than once.	5238 (87.8)	574 (9.6)	1912 (86.2)	216 (9.7)	3326 (88.8)	358 (9.6)	8.54	0.004
It is possible to contract a sexually transmitted infection without sexual intercourse.	3855 (64.6)	1200 (20.1)	1515 (68.3)	367 (16.5)	2340 (62.4)	833 (22.2)	20.89	<0.001
There are simple tests to find out whether people have HIV or not.	3035 (50.9)	1650 (27.7)	1125 (50.7)	536 (24.2)	1910 (51.0)	1114 (29.7)	0.04	0.85
People with AIDS can live as long as people without AIDS.	1496 (25.1)	1245 (20.9)	606 (27.3)	411 (18.5)	890 (23.8)	834 (22.3)	9.45	0.002
A person with HIV always looks emaciated or unhealthy.	3110 (52.1)	1252 (21.0)	1142 (51.5)	441 (19.9)	1968 (52.5)	811 (21.6)	0.60	0.44
Identification of transmission modes of HIV/AIDS	1937(32.5)	277 (4.6)	747 (33.7)	83 (3.7)	1190 (31.8)	194 (5.2)	2.34	0.13

**Table 4 ijerph-17-06716-t004:** Factors associations with sex-related knowledge among Chinese university students.

Variables	*N*	Score (M ± SD)	Correct (%)	*T* or *F*	*p* Value
Gender					
Male	2218	6.27 ± 2.62	52.3	2.71	0.007
Female	3747	6.09 ± 2.49	50.8		
Grade					
1st Year	1583	5.95 ± 2.58	49.6	9.26	<0.001
2nd Year	1753	6.08 ± 2.57	50.7		
3rd Year	1412	6.33 ± 2.50	52.8		
4th–5th Year	1217	6.38 ± 2.47	53.2		
Location of Universities					
Yunnan Province	1394	5.84 ± 2.46	48.7	231.63	<0.001
Henan Province	2965	5.71 ± 2.52	47.6		
Zhejiang Province	1606	7.29 ± 2.31	60.8		
School Category					
Top-Tier Universities	2352	6.70 ± 2.53	55.8	104.01	<0.001
First-Tier Universities	2537	5.95 ± 2.46	49.6		
Second-Tier Universities	1076	5.48 ± 2.53	45.7		
Majors					
Science	2524	6.39 ± 2.61	53.3	5.98	<0.001
Arts	3441	5.99 ± 2.48	49.9		
Childhood Residence					
Urban	2847	6.58 ± 2.54	54.8	12.24	<0.001
Rural	3118	5.78 ± 2.49	48.2		
Sexual Orientation					
Heterosexual	5196	6.11 ± 2.50	50.9	14.19	<0.001
Homosexual	158	6.33 ± 2.80	52.8		
Bisexual	287	7.10 ± 2.59	59.2		
Unclear	324	6.06 ± 2.85	50.5		
Had Sexual Intercourse					
Yes	1118	6.74 ± 2.27	56.2	8.51	<0.001
No	4847	6.03 ± 2.58	50.3		

**Table 5 ijerph-17-06716-t005:** Sexual attitudes of Chinese university students by gender.

Agreement with the Following Statements	Total (%)	Male (%)	Female (%)	χ^2^	*p* Value
Sexual Attitudes (Personal)					
It’s all right for boys and girls to kiss, hug and touch each other.	3840 (64.4)	1406 (63.4)	2434 (65.0)	1.49	0.22
There is nothing wrong with unmarried boys and girls having sexual intercourse.	2659 (44.6)	1118 (50.4)	1541 (41.1)	48.56	<0.001
A boy and a girl should have sex before they marry to see whether they are suited to each other.	1278 (21.4)	551 (24.8)	727 (19.4)	24.49	<0.001
It is all right for boys and girls to have sex with each other provided that they use methods to stop pregnancy.	2373 (39.8)	998 (45.0)	1375 (36.7)	40.06	<0.001
One-night stands are acceptable.	868 (14.6)	521 (23.5)	347 (9.3)	226.86	<0.001
A boy will not respect a girl who agrees to have sex with him.	703 (11.8)	229 (10.3)	474 (12.7)	7.25	0.007
A girl will not respect a boy who agrees to have sex with her.	485 (8.1)	201 (9.1)	284 (7.6)	4.10	0.043
Girls should remain virgins until they marry.	1423 (23.9)	550 (24.8)	873 (23.3)	1.72	0.19
Boys should remain virgins until they marry.	1410 (23.6)	496 (22.4)	914 (24.4)	3.18	0.074
Men need sex more frequently than do women.	1725 (28.9)	653 (29.4)	1072 (28.6)	0.47	0.49
It is sometimes justifiable for a boy to force a girl to have sex.	295 (4.9)	194 (8.7)	101 (2.7)	108.52	<0.001
I think that you should be in love with someone before having sex with them.	3867 (64.8)	1316 (59.3)	2551 (68.1)	46.77	<0.001
I would refuse to have sex with someone who is not prepared to use a condom.	3230 (54.1)	934 (42.1)	2296 (61.3)	206.13	<0.001
It is mainly the woman’s responsibility to ensure that contraception is used.	633 (10.6)	304 (13.7)	329 (8.8)	35.64	<0.001
I would never contemplate having an abortion myself, or for my partner.	2481 (41.6)	1007 (45.4)	1474 (39.3)	21.08	<0.001
Sexual Attitudes (General)					
Prostitution should be legalized.	540 (9.1)	286 (12.9)	254 (6.8)	63.29	<0.001
Homosexual behavior is an acceptable variation in sexual orientation.	3189 (53.5)	884 (39.9)	2305 (61.5)	262.73	<0.001
Abortion should be made available whenever a woman feels it would be the best decision.	1434 (24.0)	404 (18.2)	1030 (27.5)	65.62	<0.001
Access to pornography should be restricted among young people under the age of 18.	3957 (66.3)	1397 (63.0)	2560 (68.3)	17.77	<0.001
A person who catches a sexually transmitted disease is probably getting exactly what he/she deserves.	755 (12.7)	432 (19.5)	323 (8.6)	148.55	<0.001
A person’s sexual behavior is his/her own business, and nobody should make value judgments about it.	2790 (46.8)	961 (43.3)	1829 (48.8)	16.84	<0.001
Parents should be informed if their children under the age of eighteen have visited a clinic to obtain contraception.	4930 (82.6)	1717 (77.4)	3213 (85.7)	67.52	<0.001

**Table 6 ijerph-17-06716-t006:** Sexual behavior of Chinese university students by gender.

Variables	Total (%)	Male (%)	Female (%)	χ^2^	*p* Value
Ever had Sexual Intercourse (*n* = 5965)		*n* = 2218	*n* = 3747		
Yes	1118 (18.7)	598 (27.0)	520 (13.9)	156.60	<0.001
No	4847 (81.3)	1620 (73.0)	3227 (86.1)		
Number of People have Sex with (*n* = 1118)					
1	718 (64.2)	348 (58.2)	370 (71.2)	20.33	<0.001
≥2	400 (35.8)	250 (41.8)	150 (28.8)		
Age of First Sexual Intercourse (*n* = 1118)					
<18	171 (15.3)	109 (18.2)	62 (11.9)	8.53	0.004
≥18	947 (84.7)	489 (81.8)	458 (88.1)		
Reasons for the First Sexual Intercourse (*n* = 1118)					
I forced her/him to have intercourse against her/his will.	22 (2.0)	16 (2.7)	6 (1.2)	146.04	<0.001
I persuaded her/him to have intercourse.	92 (8.2)	89 (14.9)	3 (0.6)		
We were both equally willing.	854 (76.4)	465 (77.8)	389 (74.8)		
She/he persuaded me to have intercourse.	123 (11.0)	22 (3.7)	101 (19.4)		
She/he forced me to have intercourse.	27 (2.4)	6 (1.0)	21 (4.0)		
Using contraception during first sexual intercourse (*n* = 1118)					
Yes	826 (73.9)	442 (73·9)	384 (73.8)	<0.001	0.98
No	292 (26.1)	156 (26.1)	136 (26.2)		
Reasons for no contraception (*n* = 292)					
She/he was not willing.	19 (6.5)	4 (2.6)	15 (11.0)	14.05	0.015
I was not willing.	12 (4.1)	7 (4.5)	5 (3.7)		
Neither side is willing.	13 (1.2)	9 (5.8)	4 (2.9)		
Do not know how to do it.	30 (10.3)	20 (12.8)	10 (7.4)		
No contraceptives were prepared.	190 (65.1)	105 (67.3)	85 (62.5)		
Others	28 (9.6)	11 (7.3)	17 (12.5)		
Have/had a boyfriend or a girlfriend (*n* = 5965)					
Yes	3344 (56.1)	1315 (59.3)	2029 (54.1)	14.93	<0.001
No	2621 (43.9)	903 (40.7)	1718 (45.9)		
Ever number of boyfriend or girlfriend (*n* = 3344)					
1	1580 (47.2)	589 (44.8)	991 (48.8)	5.25	0.022
≥2	1764 (52.8)	726 (55.2)	1038 (51.2)		
Ever number of having sexual intercourse with boyfriend or girlfriend (*n* = 3344)					
0	2283 (68.3)	754 (57·3)	1529 (75.4)	128.13	<0.001
1	731 (21.9)	365 (27.8)	366 (18.0)		
≥2	330 (9.9)	196 (14.9)	134 (6.6)		
In a relationship now (*n* = 5965)					
Yes	1889 (31.7)	717 (32.3)	1172 (31.3)	0.71	0.40
No	4076 (68.3)	1501 (67.7)	2575 (68.7)		
Attitude towards current relationship (*n* = 1889)					
Casual	126 (6.7)	50 (7.0)	76 (6.5)	20.43	<0.001
Serious	897 (47.5)	324 (45.2)	573 (48.9)		
Important/might lead to marriage	768 (40.7)	285 (39.7)	483 (41.2)		
Planning to Marry	98 (5.2)	58 (8.1)	40 (3.4)		
Had sexual intercourse with current boyfriend or girlfriend (*n* = 1889)					
Yes	777 (41.1)	363 (50.6)	414 (35.3)	43.03	<0.001
No	1112 (58.9)	354 (49.4)	758 (64.7)		
Pregnancy or made Partner Pregnancy (*n* = 777)					
Yes	93 (12.0)	59 (16.3)	34 (8.2)	11.87	0.001
No	684 (88.0)	304 (83.7)	380 (91.8)		
Outcome of Pregnancy (*n* = 93)					
Currently Pregnant	19 (20.4)	14 (23.7)	5 (14.7)	18.18	0.003
Abortion	39 (41.9)	16 (27.1)	23 (67.6)		
Miscarriage	9 (9.7)	6 (10.2)	3 (8.8)		
Live-Birth	9 (9.7)	9 (15.3)	0 (0.0)		
Still Birth	2 (2.2)	1 (0.7)	1 (2.9)		
Not Sure	15 (16.1)	13 (22.0)	2 (5.9)		

**Table 7 ijerph-17-06716-t007:** Factors associated with ever having sexual intercourse among Chinese university students.

Variables	N (%)	Crude OR (95%CI)	*p* Value	Adjusted OR (95%CI)	*p* Value
Gender					
Female	520 (13.9)	1.00 (ref)		1.00 (Ref)	
Male	598 (27.0)	2.29 (2.01–2.61)	<0.001	2.38 (2.07–2.74)	<0.001
Grade					
1st Year	217 (13.7)	1.00 (ref)		1.00 (ref)	
2nd Year	266 (15.2)	1.13 (0.93–1.37)	0.23	1.34 (1.10–1.64)	0.004
3rd Year	309 (21.9)	1.76 (1.46–2.13)	<0.001	2.18 (1.79–2.66)	<0.001
4th–5th Year	326 (26.8)	2.72 (1.49–4.98)	<0.001	2.70 (2.22–3.30)	<0.001
Location of Universities					
Zhejiang Province	235 (14.6)	1.00 (ref)		1.00 (Ref)	
Yunnan Province	625 (21.1)	1.33 (1.09–1.61)	0.004	1.38 (1.13–1.69)	0.001
Henan Province	258 (18.5)	1.56 (1.32–1.84)	<0.001	1.50 (1.26–1.78)	<0.001
School Category					
Top-Tier Universities	440 (18.7)	1.00 (ref)		1.00 (ref)	
First-tier universities	436 (17.2)	0.90 (0.78–1.04)	0.17	1.04 (0.89–1.22)	0.59
Second-Tier Universities	242 (22.5)	1.26 (1.06–1.51)	0.010	1.25 (1.04–1.50)	0.020
Childhood Residence					
Rural	507 (16.3)	1.00 (ref)		1.00 (ref)	
Urban	611 (21.5)	1.41 (1.24–1.60)	<0.001	1.42 (1.23–1.63)	<0.001
Sexual orientation					
Heterosexual	972 (18.7)	1.00 (ref)		1.00 (ref)	
Homosexual	54 (34.2)	2.26 (1.61–3.16)	<0.001	2.12 (1.50–3.01)	<0.001
Bisexual	72 (25.1)	1.46 (1.10–1.92)	0.008	1.65 (1.24–2.20)	0.007
Unclear	20 (6.2)	0.29 (0.18–0.45)	<0.001	0.33 (0.21–0.53)	<0.001

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
