# Peer review of "Sexual Knowledge, Attitudes and Behaviours among Undergraduate Students in China—Implications for Sex Education"

_ijerph, 2020, doi:10.3390/ijerph17186716_

Round 1

Reviewer 1 Report

The research reported in this article is of great use to those studying issues relating to the impact of sex education on young people. I recommend that authors review some of these sources to expand and complement their critical apparatus. Some of the findings have similarities to Ibero-American research that should be reviewed to enrich the discussion, including aspects of gender identity that influence young people's sexual education and behavior.

https://www.redalyc.org/articulo.oa?id=459545417003 

https://www.redalyc.org/articulo.oa?id=368444994014

https://www.redalyc.org/articulo.oa?id=551757333002

https://www.redalyc.org/articulo.oa?id=35641005015

https://www.redalyc.org/jatsRepo/1941/194140994015/index.html

https://www.amegh.com.mx/2017/10/07/libro-a-look-into-masculine-identity-in-mexican-young-men/ 

Author Response

Response to Reviewer

We very much appreciate the constructive comments of the reviewer. We have responded point-by-point below.

Response to Reviewer 1 comments

Point 1: The research reported in this article is of great use to those studying issues relating to the impact of sex education on young people. I recommend that authors review some of these sources to expand and complement their critical apparatus. Some of the findings have similarities to Ibero-American research that should be reviewed to enrich the discussion, including aspects of gender identity that influence young people's sexual education and behavior.

https://www.redalyc.org/articulo.oa?id=459545417003 

https://www.redalyc.org/articulo.oa?id=368444994014

https://www.redalyc.org/articulo.oa?id=551757333002

https://www.redalyc.org/articulo.oa?id=35641005015

https://www.redalyc.org/jatsRepo/1941/194140994015/index.html

https://www.amegh.com.mx/2017/10/07/libro-a-look-into-masculine-identity-in-mexican-young-men/ 

Response 1: We have looked up all the links you gave us, only one of them is in English, the others are in Spanish. Since none of us are literate in Spanish, we have only quoted the English book, on page 11, lines 241-242.

Reviewer 2 Report

The study is aimed to explore gender differences in sexual knowledge, sexual attitudes and sexual behaviours as well as sex education among undergraduates in China, which shows interesting eyesight on sexual education of undergraduates. Research methodology as qustionaire is implemented in the study and the surveyed undergraduates number is sufficient. Through the statistical study general results are shown in different perspectives of sexual situation among students. They are detailed and enlightening to Chinese university educators. The conclusion that China needs to develop and widely disseminate on-line sex education with practical, age-appropriate content is urgently needed to university administrators.

Suggestions & Improveness:

  1. Literature review will be improved by classifying similar studies and showing similarities and differences between this study and others' study.
  2. The seven tables from table 1 to table 7 will be added with logic relationship in order to avoid isolation and separateness among them.
  3. The authors are expected to give specific strategies in sex education which is not found in the conclusion.

However the writing purpose is to provide specific strategies in sex education which is not found in the conclusion.

Author Response

Response to Reviewer

We very much appreciate the constructive comments of the reviewer. We have responded point-by-point below.

Response to Reviewer 2 comments

Point1 Literature review will be improved by classifying similar studies and showing similarities and differences between this study and others' study.

Response 1: The similarities and differences between this study and other studies have been added. (1) The similarities and differences between the sample size source of this study and other studies were on page 2-3, lines 89-95; (2) The similarities and differences in the incidence of sexual behaviour were on page 10, lines 218-221; (3) The similarities and differences of sexual attitudes were on page 11, lines 245-251; (4) The similarities and differences of sexual knowledge were on page 11, lines 271-272.

Point 2: The seven tables from table 1 to table 7 will be added with logic relationship in order to avoid isolation and separateness among them.

Response 2: The logical relationships among the seven tables were listed according to the sexual knowledge, sexual attitudes and sexual behavior of university students. The logical relationship is as follows: the first two are basic information, the third and fourth are sexual knowledge and its influencing factors, the fifth is sexual attitudes, and the sixth and seventh are sexual behavior and its influencing factors. We have modified some text in the tables to ensure a more logical flow.

Point 3: The authors are expected to give specific strategies in sex education which is not found in the conclusion.

Response 3: The strategy is to focus on producing and disseminating age-appropriate, high quality, accessible on-line sources for sexual education. And we added that the gender-sensitive approach to sex education should be emphasized, with a focus on empowering girls, on page 12, line 312-313.

Reviewer 3 Report

This paper is well-written and addresses important issues of young people’s sexual knowledge, attitudes, and behaviors in China. However, revisions are needed to strengthen this paper.

In the Introduction, a clearer connection needs to be made between gender equality and the need for sex education.

In general, more literature review is needed on attitudes toward sexual minorities in China. Review is also needed of prior research which addresses associations of modernization and globalization with sexual knowledge, attitudes and behaviors and attitudes toward sexual minorities.

The research questions need major revision. First, they need to be explicitly stated. Second, if sex education is part of the focus of this study, it should be included in research questions and hypotheses. Third, the first hypothesis is written in causal language. This needs to be changed as the study is cross-sectional.  Also, this study does not measure modernization and exposure to globalizing influences, which therefore should not be part of a hypothesis. The second hypothesis uses language “that gender gaps persist in sexual knowledge, attitudes and behaviors.” This hypothesis would need to be justified be a more thorough description of existing gender gaps in these areas.  It also requires a direct comparison with documented existing gender gaps, which does not seem to be part of the analysis. These research questions need major revisions.

Explanation is needed for why the specific sample was chosen, both regionally and by type of school.

The informed consent process needs documentation.

The measures section would be stronger with documentation of reliability of study measures.

The Results section should be organized by research question and should directly focus on the questions asked and hypotheses provided.

The authors do a good job in the Discussion of connecting the findings from this study with existing research. It would be strengthened by a stronger focus on the research questions, which should be consistent and clear throughout the manuscript. The authors should be careful not to overstate their findings, such as stating that their study explains lack of contraception use, which is not adequately assessed to make this claim (lines 219-222).

Author Response

Response to Reviewer

We very much appreciate the constructive comments of the reviewer. We have responded point-by-point below.

Response to Reviewer 3 comments

Point 1: In the Introduction, a clearer connection needs to be made between gender equality and the need for sex education.

Response 1: The clearer connection between gender equality and the need for sex education was added to the introduction section page2, lines 65-66.

Point 2: In general, more literature review is needed on attitudes toward sexual minorities in China. Review is also needed of prior research which addresses associations of modernization and globalization with sexual knowledge, attitudes and behaviors and attitudes toward sexual minorities.

Response 2: The attitudes toward sexual minorities in China are added to page 2, lines 52-54. The prior research which addresses associations of modernization and globalization with sexual knowledge, attitudes and behaviors and attitudes toward sexual minorities was added to page 2, lines 50-55 and page11, line 257-263.

Point 3: The research questions need major revision. First, they need to be explicitly stated. Second, if sex education is part of the focus of this study, it should be included in research questions and hypotheses. Third, the first hypothesis is written in causal language. This needs to be changed as the study is cross-sectional.  Also, this study does not measure modernization and exposure to globalizing influences, which therefore should not be part of a hypothesis. The second hypothesis uses language “that gender gaps persist in sexual knowledge, attitudes and behaviors.” This hypothesis would need to be justified be a more thorough description of existing gender gaps in these areas.  It also requires a direct comparison with documented existing gender gaps, which does not seem to be part of the analysis. These research questions need major revisions.

Response 3: Based on your comments, we have revised the research question section and removed the hypotheses. Based on the existing results and analysis, we deleted the original hypotheses 1 and 2 and replaced them with 3 research questions, on page 2, lines 80-83. This study does not measure modernization and exposure to globalizing influences, so we removed the hypothesis 1. Hypothesis 2 was replaced with research question 2 " the gender differences in sexual attitudes and sexual behaviours ", which was more in line with the analysis in the results and added more discussion, on page 11, lines 238-245. And according to the research questions, the discussion, conclusion and abstract were adjusted to make the whole article more suitable for the research questions.

Point 4: Explanation is needed for why the specific sample was chosen, both regionally and by type of school.

Response 4: The three provinces of Zhejiang, Henan, and Yunnan are chosen because they not only represent the three regions of China's east, middle, and west, but also represent high, medium and low economic levels. In addition few previous studies have explored students at different university levels. We added it to page 2-3, lines 89-95.

Point 5: The informed consent process needs documentation.

Response 5: The informed Consent (in Chinese) of this study has been attached at the end of this document for your review.

Point 6: The measures section would be stronger with documentation of reliability of study measures.

Response 6: In the Attitudes Towards Sexuality Scale, we have added internal reliability and test-retest reliability, on page 3, line 116-117. Illustrative Questionnaire for Interview­ Surveys with Young People (IQISYP) comes from the World Health Organization. It is an interview questionnaire, not suitable for reliability analysis, but is aimed specifically at exploring sexual behavior in young people in this age range.

Point 7: The Results section should be organized by research question and should directly focus on the questions asked and hypotheses provided.

Response 7: We have revised the research question, on page 2, lines 80-83, to make the research questions more consistent.

Point 8: The authors do a good job in the Discussion of connecting the findings from this study with existing research. It would be strengthened by a stronger focus on the research questions, which should be consistent and clear throughout the manuscript. The authors should be careful not to overstate their findings, such as stating that their study explains lack of contraception use, which is not adequately assessed to make this claim (lines 219-222).

Response 8: We changed the order of discussing the third and fourth points, and revised the content of the new fourth point to focus on the research questions. We removed the statement about lack of contraception from the manuscript. On page 11, lines 231-233.

Round 2

Reviewer 2 Report

  1. It is good to improve your paper, which is more clarified to readers.
  2. Literature review can be improved to introduce other scholars' research in the field of Sexual Knowledge , Attitudes and Behaviours in recent five years.
  3. It is better to show similarities and differences between your study and others in the first part.

Author Response

Response to Reviewer

We very much appreciate the constructive comments of the reviewer. We have responded point-by-point below.

Response to Reviewer 2 comments

Point 1: Literature review can be improved to introduce other scholars' research in the field of Sexual Knowledge, Attitudes and Behaviours in recent five years.

Response 1: The other scholars' research in the field of sexual knowledge, attitudes and behaviours, including recent studies, has been added to page 2, line 58-78.

Point 2: It is better to show similarities and differences between your study and others in the first part.

Response 2: The similarities and differences between this study and other studies have been added to page 3, line 112-115 and line 124-129.

Reviewer 3 Report

Overall, the authors have improved the manuscript. However, there are issues that should be addressed before it is accepted for publication. The authors still need to explain clearly how sex education supports gender equity. The research questions are much improved, focusing on measurable questions that are assessed in this study. However, the issues raised in two of the research questions need to be addressed more thoroughly in the literature review. First, a review of research is needed on Chinese university students’ sexual knowledge and attitudes. Second, the question about gender differences in sexual knowledge behavior needs to be preceded by a review of existing research on this area. Also, how will this question help to understand gender equity issues, which seems to be a primary concern raised in the introduction? Further, there needs to be a clearer connection between the information in the introduction about homosexuality and the research questions. Perhaps the authors can add this focus to the first research question, since it is a key aspect of study findings.

Author Response

Response to Reviewer

We very much appreciate the constructive comments of the reviewer. We have responded point-by-point below.

Response to Reviewer 3 comments              

Point 1: The authors still need to explain clearly how sex education supports gender equity.

Response 1: Explanation about how sex education supports gender equity has been added to the introduction section page 3, line 105-110 and the discussion section page 13, line 336-344.

Point 2: First, a review of research is needed on Chinese university students’ sexual knowledge and attitudes.

Response 2: The review about Chinese university students’ sexual knowledge, attitudes and behaviour was added to page 2, line 56-78.

Pint 3: Second, the question about gender differences in sexual knowledge behavior needs to be preceded by a review of existing research on this area. Also, how will this question help to understand gender equity issues, which seems to be a primary concern raised in the introduction?

Response 3: The gender differences in sexual knowledge and behavior were added to page 2, 89-93.

Point 4: Further, there needs to be a clearer connection between the information in the introduction about homosexuality and the research questions. Perhaps the authors can add this focus to the first research question, since it is a key aspect of study findings.

Response 4: A clearer connection between the information in the introduction about homosexuality and the research questions was added to page 2, line 74-78. We added the LGBT students to the first research question.
